# Adjusting Haemoglobin Values for Altitude Maximizes Combined Sensitivity and Specificity to Detect Iron Deficiency among Women of Reproductive Age in Johannesburg, South Africa

**DOI:** 10.3390/nu12030633

**Published:** 2020-02-27

**Authors:** Takana Mary Silubonde, Jeannine Baumgartner, Lisa Jayne Ware, Linda Malan, Cornelius Mattheus Smuts, Shane Norris

**Affiliations:** 1Centre of Excellence for Nutrition, Faculty of Health Sciences, North-West University, Potchefstroom 2531, South Africa; 2SAMRC/Wits Developmental Pathways for Health Research Unit, Faculty of Health Sciences, University of the Witwatersrand, Johannesburg 2000, South Africa; 3Laboratory of Human Nutrition, Department of Health Sciences and Technology, ETH Zurich, 8092 Zurich, Switzerland; 4School of Health and Human Development, University of Southampton, Southampton S016 6YD, UK

**Keywords:** anaemia, altitude adjustment, haemoglobin, iron, South Africa, women of reproductive age

## Abstract

In South Africa, haemoglobin (Hb) is measured to screen for iron deficiency (ID). However, low levels of Hb are only a late stage indicator of ID. Furthermore, Hb values are generally not adjusted for altitude even though recommended by WHO. We determined the Hb threshold with the highest combined sensitivity and specificity for detecting ID among South African women living at 1700 m above sea level. In a cross-sectional study of 492 18–25-year-old women, we measured Hb and iron status biomarkers. Using receiver operating characteristic curves, we determined the Hb threshold with maximum Youden Index for detecting ID. This threshold of <12.35 g/dL resulted in a 37.2% anaemia prevalence (20.9% IDA), and sensitivity and specificity of 55.7% and 73.9%, respectively. The WHO altitude-adjusted threshold of <12.5 g/dL resulted in a 39% anaemia prevalence (21.3% IDA), and sensitivity and specificity of 56.8% and 70.8%, respectively. In contrast, using the unadjusted Hb cut-off of <12 g/dL resulted in a 18.5% anaemia prevalence (12.6% IDA), and sensitivity and specificity of 35.1% and 88.6%, respectively. In this sample of South African women of reproductive age an Hb threshold <12.35 g/dL had the highest combined sensitivity and specificity for detecting ID. The diagnostic performance of this Receiver operating characteristic curve-determined threshold was comparable to the altitude-adjusted threshold proposed by WHO. Thus, clinical and public health practice in South Africa should adopt adjustment of Hb for altitude to avoid underestimation of ID and missing women in need for intervention.

## 1. Introduction

Iron deficiency (ID) is the most common micronutrient deficiency worldwide and the major cause of anaemia. Women of reproductive age (WRA) have high iron requirements because of iron loss through menstruation. An estimated half a billion WRA are affected by anaemia, with ID being responsible for approximately half the cases [1]. During pregnancy, physiological iron requirements increase even further to ensure adequate blood volume expansion and optimal placental and foetal development [2]. However, in low-and middle-income countries, iron intake and/or absorption are often inadequate to meet demands, resulting in ID and ID anaemia (IDA) [3].

Maternal ID and IDA are associated with serious consequences for the mother and her offspring, including an increased risk of morbidity and mortality, adverse birth outcomes, and impaired physical and neurological development in the offspring [4]. In an effort to prevent the adverse health effects of ID and IDA, the World Health Organization (WHO) recommends intermittent iron-folic acid supplementation in menstruating women living in settings where the prevalence of anaemia is 20% or higher [5], and daily oral iron-folic acid supplementation in all pregnant women as part of routine antenatal care [6]. To identify populations at risk for ID and individuals in need for treatment, accurate diagnosis is imperative.

In primary health care settings, the measurement of haemoglobin (Hb) concentration is commonly used to screen individuals who require iron supplementation, e.g., by using inexpensive and easy to use point-of-care haemoglobinometers [7]. Hb is an iron-containing protein found within red blood cells (RBCs). Mature human RBCs have a life span of ~120 days, after which they become senescent and are phagocytosed by macrophages, with iron being recycled [8]. Thus, Hb concentrations only drop when iron stores are depleted (the late stages of ID) or when iron cannot be mobilized from storage in hepatocytes or macrophages, e.g., in the presence of inflammation [9,10]. Furthermore, anaemia has a multifactorial aetiology and can exist without ID [11]. Therefore, measuring Hb concentrations has generally low sensitivity (ability of a test to correctly identify those with the condition) and specificity (ability of the test to correctly identify those without the condition) for the detection of ID [12,13]. Nonetheless, due to the lack of a single, inexpensive, and simple measure of iron status, measuring Hb concentration remains the method of choice for ID screening in primary healthcare, particularly in low-resource settings.

However, the interpretation of Hb values is not free of challenges. Hb cut-off points to diagnose anaemia remained relatively unchanged since 1968, and have been defined for different age categories of children, men, non-pregnant and pregnant women [12]. Since 2001, the WHO further recommended adjusting Hb concentrations downwards (or correcting cut-off point upwards) in individuals residing at altitudes higher than 1000 m above sea level [12]. Erythropoiesis (production of RBCs) increases as a result of chronic hypoxia [14]. Partial pressure of oxygen decreases as altitude increases, resulting in a lower oxygen saturation of RBCs and an increase in erythropoiesis as an adaptive response [15]. Thus, failure to adjust Hb values for altitude may lead to an underestimation of the anaemia and IDA prevalence at population level, and missed diagnosis and subsequent treatment at an individual level.

Despite these risks, the adjustment of Hb values for altitude is not a standard practice, neither by researchers nor by health care professionals in clinical settings, and remains a matter of debate [16,17]. Sharma and colleagues recently re-examined associations of Hb with altitude in a dataset including 68,193 observations among preschool-aged children and WRA from all WHO regions (except South-East Asia) [18]. The authors confirmed that Hb should be adjusted for altitude, but indicated that current recommendations may underestimate anaemia for those residing at lower altitudes (<2000 m) and overestimate anaemia for those residing at higher altitudes (<3000 m). Furthermore, small differences in Hb concentrations may exist between different ethnic groups. Thus, Karakochuk et al. recently stated that the validity of adjusting Hb cut-offs in individuals of African origin warrants further research [7].

South Africa occupies the southern tip of Africa, and consists of a high-lying inland plateau separated from the narrow, low-lying coastal plain by mountainous escarpment. The country’s mean altitude is approximately 1200 m, and at least 40% of the surface is at a higher elevation [19]. Nonetheless, South African clinical guidelines do not advise adjusting Hb values [20,21,22], which is therefore not a general practice in the public healthcare sector. However, in the most recent South African Demographic and Health Survey (SADHS) conducted in 2016 [23], anaemia prevalence rates were reported based on altitude-adjusted Hb values, while previous national surveys, such as the 2012 South African National Health and Nutrition Examination Survey (SANHANES) [24], reported unadjusted values. It can be speculated that this may explain the increase in anaemia prevalence in WRA from 23.1% to 31.5% in the 2012 and 2016 surveys, respectively. Consensus is needed on whether to adjust Hb values for altitude or not in South African population surveys and in the primary health care sector.

Therefore, the aim of this study was to establish an Hb threshold (≈adjustment value) that would maximize the combined sensitivity and specificity for the detection of ID among WRA residing in Soweto, South Africa, situated at 1700 m above sea level. Our objectives were three-fold: (i) to determine the Hb cut-off point with the highest combined sensitivity and specificity (maximum Youden Index) for detecting ID based on ferritin [adjusted and non-adjusted for inflammation] and iron deficient erythropoiesis (IDE) based on soluble transferrin receptor (sTfR); (ii) to assess and compare the sensitivity and specificity of the Hb cut-off points adjusted for altitude (as recommended by WHO) and without adjustment for the detection of ID; and (iii) to determine and compare the prevalence of anaemia, IDA and ID without anaemia using these different cut-off points.

## 2. Materials and Methods

### 2.1. Study Setting and Population

For this study, we used data collected between 2018 and 2019 from the *Soweto Young Women’s Survey*, a prospective study conducted in Gauteng, South Africa’s most populous province (home to approximately 26% of the South African population). Data collection took place at the South African Medical Research Medical Council (SAMRC) Developmental Pathways to Health Research Unit (DPHRU), located within the Chris Hani Baragwanath Academic Hospital (CHBAH), a tertiary hospital in Soweto (South Western Township). Generally healthy, 18–25-year-old, non-pregnant WRA were recruited from Soweto, a historically disadvantaged urban area of 200 km^2^ in the city of Johannesburg, Gauteng province. Over 1.3 million people reside in Soweto (6400/km^2^) with 98.5% of the population being of black African descent. Women were eligible for inclusion if they were aged 18–25 years; proficient in local languages; born in South Africa or neighbouring countries; and if they had been residing in their home in Soweto for at least 3 months. Women were excluded if they had a previous diagnosis of type-1 diabetes; cancer or epilepsy; or not able or willing to provide written informed consent. Due to South Africa’s high prevalence of HIV infection (23.2% of women aged 15–49 years [25]), women who were HIV positive were included in the study for the sample to be a better representation of the general population. As a result, HIV was self-reported and CD4 and viral load were not assessed.

A cluster design was employed for recruitment, where each Soweto community centre was a cluster. Thirty clusters with a radius of 1 km^2^ each were identified around Soweto using churches as the midpoint of each cluster. An online search was performed using the Google search engine to locate the information of all churches in Soweto. Using street address information, geolocations of each church structure were obtained, and each church was visited by fieldworkers and verified. The latitude and longitude of the 104 churches identified and verified were then classified using k-means clustering. The church with the shortest straight line distance to the cluster centroid was selected for inclusion in the study as it was at the centre of a cluster of churches that was maximally distant from the other churches in Soweto. An equal number of participants were recruited from two randomly selected clusters.

During the recruitment process, all households in the selected community clusters were approached to identify those where young women lived. Once identified, potential participants were visited in their homes and were informed in their home language about: (i) the objectives of the study; (ii) the use of the results; and (iii) the risks and benefits of the study. An informed consent form was supplied to potentially eligible women who were interested in being part of the study. The interested women were then invited to the study site for the signing of informed consent and data collection.

### 2.2. Ethical Considerations

This study was conducted in accordance with the ethical principles laid down in the Declaration of Helsinki, and all procedures involving human participants were approved by the Human Research Ethics Committees of the North-West University, Potchefstroom (NWU-0042919-S1), and the University of the Witwatersrand, Johannesburg (M171137).

### 2.3. Measurement of Anthropometric Indicators and Socio-Demographic Information

Weight (kg) and height (cm) of participants were measured in order to calculate body mass index: BMI = weight (kg)/(height [m])^2^. Weight was measured using the Seca 877 Scale (Seca, Hamburg, Germany) and recorded to the nearest 100 g. Women wore light clothing and removed shoes and heavy outerwear (e.g., sweaters) before obtaining weight. Height was measured to the nearest 0.1 cm using a single calibrated Holtain Stadiometer (Holtain Limited, Crymych, UK). Participants were measured either barefoot or wearing thin socks. Mid upper arm circumference (MUAC) was measured to the nearest 0.1 cm using a plastic measuring tape. Measurement was taken at the mid-point of the upper arm, between the acromion process and the tip of the olecranon. A MUAC ≤24 cm was used to define undernutrition [26]. A questionnaire was administered by a trained research assistant to assess socio-economic and demographic characteristics of the women. Food insecurity (hunger) was assessed by a shortened version of the Community Childhood Hunger Identification Project (CCHIP) Index [27].

### 2.4. Biomarker Analysis

Hb concentrations were measured at the health research unit in capillary blood collected by an experienced nurse using a calibrated Hb 201+ HemoCue^®^ system (HemoCue Johannesburg, South Africa). The HemoCue^®^ system is an easy-to-use haemoglobinometer that is commonly used for determining haemoglobin concentrations in health research and for point-of-care testing in primary health care settings [7,28]. The South African point-of-care cut-off to diagnose anaemia and ID in WRA is a finger-prick Hb <12 g/dL [20,22]. Participants diagnosed with anaemia, according to these national guidelines, were given a referral letter to visit their nearest clinic.

Venous blood samples were drawn into lithium heparin tubes (BD, Plymouth, UK). Plasma was separated within 1 h after blood collection, and aliquots stored at −20 °C for a maximum of 14 days until transportation (on dry ice) for storage at −80 °C until analysis. For the analysis of iron status indices and the inflammation/infection markers C-reactive protein (CRP) and alpha-1-acid glycoprotein (AGP), the Q-Plex™ Human Micronutrient Array (7-plex, Quansys Bioscience, Logan, UT, USA) was used [29].

As plasma ferritin concentrations are elevated in the presence of inflammation, ferritin concentrations were adjusted for inflammation using the correction factors proposed by Thurnham et al. [30]. For participants in the incubation phase (CRP > 5 mg/L and AGP ≤ 1 g/L), ferritin values were adjusted using a correction factor of 0.77. For the early convalescence phase (CRP > 5 mg/L and AGP > 1 g/L), a correction factor of 0.53 was used, while in the late convalescence phase (CRP ≤ 5 mg/L and AGP > 1 g/L), a correction factor of 0.75 was applied.

Participants were classified as being iron deficient if their inflammation-adjusted plasma ferritin concentration was <15 μg/L or, if their unadjusted plasma ferritin concentration was <30 µg/L as recommended by WHO in settings where infection or inflammation is prevalent [31]. However, since the ferritin cut-off of <15 μg/L is still commonly used for unadjusted ferritin values in the diagnosis of ID in the public health sector in South Africa (e.g., population surveys), we also reported the ID prevalence based on unadjusted ferritin values using a cut-off of <15 µg/L. IDA was defined as ferritin <15 μg/L (adjusted) or <30 µg/L (unadjusted) plus haemoglobin <12 g/dL. Iron deficient erythropoiesis (IDE) was defined as sTfR ≥8.3 mg/L.

### 2.5. Data and Statistical Analysis

Study data were collected and managed using REDCap electronic data capture tools hosted at the University of Witwatersrand [32]. Data processing and statistical analysis of data were performed using Microsoft Office Excel 2010 (Microsoft, Redmond, WA, USA) and SPSS version 25 (SPSS Inc., Chicago, IL, USA).

Data were tested for normality by means of visual inspection using Q-Q plots and histograms, and the Shapiro–Wilk test. Normally distributed data are expressed as means ± SD; non-normally distributed data are expressed as medians (IQR).

Receiver operating characteristic (ROC) curves were created for the use of Hb in the detection of ID defined by ferritin (inflammation-adjusted <15 µg/L; unadjusted <30 µg/L) and sTfR (≥8.3 mg/L). The diagnostic threshold of Hb indicative of ID was selected based on the Youden Index, which is defined by the formula: sensitivity + specificity − 1. The maximal Youden Index identifies the optimal threshold when sensitivity and specificity are considered of equal importance, and this maximal value is the “knee” of the ROC curve [33]. For each ROC curve, the area under the curve (AUC) is reported. An AUC value of 0.50 indicates completely random predictions while a value of 1 indicates perfect predictions.

Furthermore, we determined the sensitivity, specificity, and Youden Index for the diagnostic thresholds indicative of ID (based on inflammation-adjusted and unadjusted ferritin or cut-off values) when Hb values were adjusted for altitude (Hb value—0.5 g/dL at 1700 m above sea level), which is equivalent to the use of an Hb cut-off of <12.5 g/dL, as recommended by WHO [12].

## 3. Results

### 3.1. Sample Characteristics

The characteristics of the 492 WRA participating in this study are shown in Table 1. All women were of African descent with a median age of 21 years (IQR 19–23). The majority of the women (61%) had obtained a high school leaving certificate. The median household size was six residents (IQR 4–8) and 47% of women reported food insecurity. Half of the women (51%) were nulliparous. The median BMI was 24.4 (21.2–29.6) kg/m^2^, with 22% of women overweight and 24% obese.

### 3.2. Hematological Indicators

Table 2 presents the inflammatory and iron status indicators in the sample of WRA. Thirty-four percent of the women had inflammation/infection (CRP > 5 mg/L and/or AGP > 1 g/L) with 14% of the women in the late convalescence stage. The median adjusted ferritin concentration was 25.9 (8.0–55.1) µg/L, and the median unadjusted ferritin 28.2 (9.3–62.9) µg/L. The prevalence of ID based on the adjusted and unadjusted ferritin values was 38% and 36%, respectively, when using the cut-off point of <15 µg/L, but 52% for unadjusted ferritin values with the cut-off point of <30 µg/L. The prevalence of IDE (sTfR > 8.3 mg/L) was 42%.

Figure 1 shows receiver operating characteristic (ROC) curves for the use of Hb to diagnose ID (Figure 1a–c) and IDE (Figure 1d). ROC curve analysis resulted in an Hb threshold of 12.35 g/dL with maximum sensitivity and specificity (Youden Index = 0.30) and an AUC of 0.681 to detect ID based on an inflammation-adjusted ferritin <15 µg/L (Figure 1a). To detect ID based on an unadjusted ferritin <30 µg/L (WHO recommendation) or <15 µg/L (current clinical and public health practice), ROC curve analysis resulted in an Hb threshold of 12.45 g/dL (Figure 1b,c) and an AUC of 0.651 and 0.675, respectively. When using sTfR to define IDE, ROC curve analysis resulted in an Hb threshold of 12.35 g/dL and AUC of 0.633.

Table 3 shows the diagnostic accuracy of the Hb cut-off points to detect ID based on inflammation-adjusted and unadjusted ferritin values, and presents the iron and anaemia status of the participating WRA based on the different Hb cut-off points and ferritin criteria to define ID. The unadjusted Hb cut-off point of <12.00 g/dL, that is currently used in primary health care, had a sensitivity of 35.1% and specificity of 88.6% to diagnose ID based on an inflammation-adjusted ferritin <15 µg/L. It resulted in an anaemia, IDA and ID without anaemia prevalence of 18.5% and 12.6% and 25.0%, respectively. The ROC-curve-determined Hb cut-off point of 12.35 g/dL optimized to diagnose ID based on an inflammation-adjusted ferritin <15 µg/L had a sensitivity of 55.7% and specificity of 73.9%. It resulted in an anaemia, IDA and ID without anaemia prevalence of 37.2% and 20.9% and 16.7%, respectively. The ROC-curve-determined Hb cut-off point of 12.45 g/dL to determine ID based on an unadjusted ferritin <30 µg/L had a sensitivity of 51.6% and a specificity of 74.4%. It resulted in an anaemia, IDA and ID without anaemia prevalence of 39.0% and 26.6% and 25.0%, respectively. The ROC-curve-determined Hb cut-off point of 12.45 g/dL to determine ID based on an unadjusted ferritin <15 µg/L had a sensitivity of 56.2% and a specificity of 70.7%, and resulted in an anaemia, IDA and ID without anaemia prevalence of 39.0% and 20.3% and 14.6%, respectively. The altitude-adjusted Hb cut-off point of <12.5 g/dL, as recommended by WHO, had a sensitivity of 56.8% and a specificity of 70.8%. It resulted in an anaemia, IDA and ID without anaemia prevalence of 39.0% and 21.3% and 16.3%, respectively.

## 4. Discussion

The results of this analysis among 18–25 year-old South African WRA living at 1700 m above sea level showed that the Hb threshold with the highest combined sensitivity and specificity for detecting both ID (based on inflammation-adjusted ferritin <15 µg/L) and IDE (based on sTfR > 8.3 mg/L) is <12.35 g/dL. The diagnostic performance of this ROC-curve-determined cut-off value was comparable to the altitude-adjusted Hb cut-off of <12.5 g/dL proposed by WHO for this altitude. In contrast, the Hb cut-off of <12.00 g/dL, which is currently used in the South African primary health care setting without adjusting Hb values for altitude, had a lower sensitivity for the detection of ID than the ROC-curve-determined and altitude-adjusted cut-offs (35% versus ~56%). According to our analysis, the current Hb threshold used in primary health care for the detection of anaemia missed more than half of the anaemia cases detected when using an altitude adjusted Hb threshold (18.5% versus 39%), and resulted in a markedly lower prevalence of IDA (12.6% versus 21%), and a higher prevalence of ID without anaemia (25% versus ~16%). The latter indicates the proportion of women with depleted iron stores who would be missed if measuring Hb to screen for ID.

According to the 2018 Standard Treatment Guidelines and Essential Medicines List for South Africa, the first step in the diagnosis of ID at primary healthcare level is measuring Hb [22]. Thereby, a Hb <12 g/dL is considered indicative of anaemia in non-pregnant WRA, which requests a full blood count to determine the likely aetiology of anaemia based on measured mean corpuscular volume (MCV; average size and volume of red blood cells). If MCV is normal, systemic disease or haemolysis are considered likely causes, and if MCV is low then ID is the most likely cause, while high MCV is indicative of a folate and/or vitamin B_12_ deficiency [22]. Only an estimated 50% of anaemia cases are attributable to ID [28], and therefore proper identification of the anaemia cause is necessary to ensure cause-specific treatment [34]. However, follow-up assessments to confirm the cause of anaemia and the presence of IDA, including the one described above, are rarely done in primary healthcare clinics in South Africa.

Furthermore, clinical guidelines do not specify how to interpret Hb values from individuals residing in higher altitude settings [20,21,22], which makes up a substantial proportion of South Africa’s population [35]. The results of this study indicate that the current point-of-care threshold to detect anaemia (<12 g/dL) has poor sensitivity (35.1%) for the detection of ID (88.6% specificity) in women residing at an altitude of 1700 m above sea level. In our sample, the ROC curve was optimized at an Hb threshold of <12.35 (AUC = 0.681) using the Youden Index, which defines the maximum potential effectiveness of Hb to detect ID when equal weight is given to sensitivity and specificity. Thereby, the ROC curve-determined threshold increased sensitivity to 55.7%, while reducing specificity to 73.9%. The diagnostic performance of this threshold was comparable to the performance of the altitude-adjusted Hb threshold (<12.5 g/dL) proposed by WHO, which slightly improved sensitivity (56.8%) while maintaining a favourable specificity (70.8%). This confirms the findings by Sharma et al., who recently re-examined haemoglobin adjustments to define anaemia among WRA residing at different altitudes and concluded that Hb values (or thresholds) should be adjusted for altitude [18].

Not only can the lack of altitude-adjustment lead to missed diagnosis and treatment of ID in primary health care clinics, it can also result in an underestimation of the anaemia burden in the population. At the public health level, WHO recommends intermittent (once, twice, or three times per week on non-consecutive days) iron and folic acid supplementation of adult women and adolescent girls in populations where the prevalence of anaemia among WRA is 20% or higher [5]. The recommendation for blanket iron supplementation emphasizes that sensitivity of Hb as a screening tool to detect ID is more important than specificity in the public health context. Nonetheless, sensitivity alone does not provide the basis for informed decisions following positive screening test results because those positive results could contain many false positive outcomes [36], hence the use of the Youden Index to define a cut-off with maximum combined sensitivity and specificity. The anaemia prevalence of 18.5% obtained in our sample when using the non-adjusted Hb cut-off of <12.00 g/dL would indicate that no intervention at the public health level is needed. In contrast, the 39% anaemia prevalence obtained when using the ROC-curve-determined and the altitude-adjusted cut-offs would indicate a moderate to borderline severe public health problem that requires intervention. Recent national surveys have reported discrepant anaemia prevalence rates among South African WRA. The (South African Demographic Health Survey) SADHS conducted in 2016 reported an anaemia prevalence in WRA of 31.5% [23], while the South African National Health and Nutrition Examination Survey (SANHANES) 2012 reported an anaemia prevalence of 23.1% [24]. The latter survey did not adjust Hb values for altitude and used the HemoCue 201+ system to measure Hb while SANHANES 2012 used an automated haematology analyser. These and other factors make it impossible to conclude whether the situation truly got worse or whether the 2012 survey potentially underestimated the anaemia prevalence [37].

Another challenge lays in the interpretation of ferritin in settings with a high prevalence of inflammation. Ferritin is not only a marker of iron stores but also an acute-phase protein that increases during inflammation independently of iron status [38,39]. In this study, 34% of women had inflammation, highlighting the importance of considering inflammation when interpreting ferritin values and determining ID prevalence. Possible strategies to interpret ferritin values in settings with a high inflammatory burden are to raise the cut-off that defines deficiency [31], or to adjust ferritin values of individuals with inflammation [30,39]. The latter requires the measurement of inflammatory markers, which is not always feasible. Furthermore, there is currently no consensus on the most appropriate adjustment method. For the determination of an optimized Hb threshold to detect ID, we defined ID using different ferritin interpretation approaches. ROC analysis using unadjusted ferritin cut-off values of either <30 or <15 µg/L to define ID resulted in an optimized Hb cut-off of <12.45 g/dL. However, the highest Youden Index and AUC for Hb to detect ID was obtained when ferritin was adjusted for inflammation using the correction factors proposed by Thurnham et al. In the current study, we used the correction factors proposed by Thurnham et al. [30]. However, for interest sake we also adjusted ferritin values using the Biomarkers Reflecting Inflammation and Nutritional Determinants of Anaemia (BRINDA) approach and repeated the ROC curve analysis for Hb to detect ID based on BRINDA-adjusted ferritin <15 µg/L [40]. This resulted in an even lower median ferritin concentration of 19.8 (6.2–41.2) µg/L and ID prevalence of 55.3%. Nonetheless, the Hb threshold with optimized combined sensitivity and specificity to detect ID remained at <12.35 g/dL (Youden Index = 0.281), with a sensitivity and specificity of 52.7% and 74.4%, respectively. Thus, overall, the different approaches to interpret ferritin did not relevantly affect the diagnostic performance of Hb to detect ID, but the use of the higher ferritin cut-off of <30 µg/L likely resulted in an overestimation of ID and IDA.

When interpreting the results of this study, the following limitations must be considered. Firstly, since we only included data from women residing at one particular altitude, we cannot draw a definite conclusion on whether the lack in sensitivity of the current point-of-care cut-off is driven by altitude or by other factors. Thus, future studies should repeat these analyses in South African women residing at different altitudes. Secondly, we did not perform a full blood count using an automated haematology analzyer to validate the accuracy of the HemoCue 201+ system and to determine the likely causes of anaemia using MCV. The HemoCue 201+ system has been used in the most recent SADHS survey (versus automated haematology analyser in SANHANES) and is being used in most primary healthcare settings in South Africa to screen for anaemia and IDA. Nonetheless, studies that compared the HemoCue 201+ system with automated haematology analyzers showed that Hb concentrations measured by the HemoCue 201+ system were on average 0.1 to 0.4 g/dL higher than when measured with the haematology analyser [7,37]. This would suggest that if measured with an automated haematology analyser, the ROC-curve-determined Hb threshold for detecting ID would have been slightly higher, and therefore getting even closer to the altitude-adjusted threshold recommended by WHO. Lastly, we did not consider the potential effects of smoking on Hb concentrations in our analysis, since information on smoking was incomplete. Similar to altitude-exposure, the reduced oxygen-carrying capacity in smokers caused by exposure to carbon monoxide results in a compensatory increase in Hb concentrations in individuals smoking ≥1/2 packet per day [40]. Therefore, WHO recommends adjustment of Hb for smoking, and Sharma et al. showed that adjustments for smoking and altitude are additive [12]. Thus, we cannot rule out that Hb concentrations of smoking WRA would have needed even further adjustment, again potentially closing the gap between the ROC-determined and the altitude-adjusted Hb threshold.

The strength of this study was the assessment of iron status by measuring two different biomarkers (ferritin and sTfR) using a validated method [29]. However, we do acknowledge that a recent study comparing the Quansys multiplex immunoassay with reference-type assays found poor precision for sTfR in comparison with the Roche cobas 6000 clinical analyser [41]. Nonetheless, the Hb threshold that we obtained by ROC curve analysis for the detection of ID (based on ferritin) was identical to the one for the detection of IDE (based on sTfR), although the latter mainly served as sensitivity analysis. In addition, the analysis of both CRP and AGP enabled us to adjust ferritin for inflammation and perform a sensitivity analysis.

## 5. Conclusions

In conclusion, the sensitivity and specificity of the optimized Hb cut-off point of 12.35 g/dL determined by ROC curve analysis for the detection of ID was comparable to the Hb cut-off point adjusted for altitude (<12.5 g/dL) as recommended by WHO. It further resulted in comparable anaemia, IDA, and ID without anaemia prevalence rates. In contrast, the use of an unadjusted Hb cut-off point (or lack of adjusting single Hb values) resulted in an underestimation of the anaemia and IDA prevalence, and missed detection of ID in 25% of women (versus 16% for altitude-adjusted cut-off) when analysis of additional iron status indicators is not possible or standard practice. Thus, this study confirms that clinical and public health practice should adopt the adjustment of Hb values for altitude according to WHO guidelines to avoid underestimation of the anaemia and IDA burden in the population and missing individuals in need for intervention. The lack of addressing a large anaemia burden in WRA may not only compromise their own health, but also the health of their baby if they become pregnant.

## Figures and Tables

**Figure 1 nutrients-12-00633-f001:**
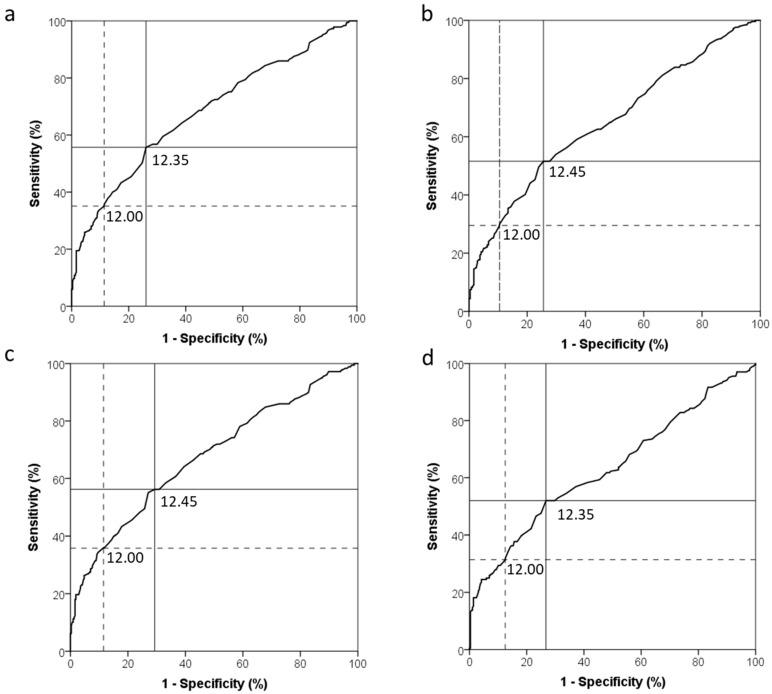
Receiver operating characteristic curves for the use of haemoglobin to diagnose iron deficiency in 18–25 year-old non-pregnant women of reproductive age (n = 492) based on: (**a**) Inflammation-adjusted ferritin (<15 µg/L), area under the curve (AUC) = 0.681; (**b**) unadjusted ferritin (<30 µg/L), AUC = 0.651; (**c**) unadjusted ferritin (<15 µg/L), AUC = 0.675; and (**d**) soluble transferrin receptor (sTfR > 8.3 mg/L), AUC = 0.633. The solid line indicates the haemoglobin cut-off point with the highest combined sensitivity and specificity (maximum Youden Index), while the dotted line indicates the unadjusted haemoglobin cut-off point, typically used in the public healthcare sector.

**Table 1 nutrients-12-00633-t001:** Characteristics of 18–25 year-old women of reproductive age (WRA) residing in Soweto, South Africa (*n* = 492).

Characteristics	Median (IQR) or *n* (%)
Age	21 (19–23)
BMI (kg/m^2^)	24.4 (21.2–29.6)
Underweight (<18.5 kg/m^2^)	41 (8)
Normal weight (18.5–24.9 kg/m^2^)	224 (46)
Overweight (25–29.9 kg/m^2^)	110 (22)
Obese (>30 kg/m^2^)	117 (24)
MUAC (cm)	27.6 (24.8–31.5)
Undernutrition (≤24 cm)	94 (19)
HIV positive (self-reported) Yes	22 (4)
Food insecurity	
Yes	230 (47)
Household asset score	9 (7–10)
Low (1–5)	38 (8)
Medium (6–9)	308 (63)
High (10–13)	146 (30)
Highest level of education	195 (40)
Primary school or less	297 (61)
High school leaving certificate	
Household size (number of people)	6 (4–8)
1–4	154 (32)
5–10	269 (56)
>10	57 (12)
Parity	
Nulliparous	253 (51)
Primiparous	192 (39)
Multiparous	47 (10)

BMI = body mass index, MUAC = mid upper arm circumference.

**Table 2 nutrients-12-00633-t002:** Inflammatory and iron status indicators in 18–25 year-old women of reproductive age residing in Soweto, South Africa (*n* = 492).

Biomarker	Median (IQR) or *n* (%)
CRP (mg/L)	1.41 (0.44–3.84)
AGP (g/L)	0.86 (0.72–1.02)
Inflammatory status	
No inflammation (CRP ≤ 5 mg/L and AGP ≤ 1 g/L)	327 (67)
Incubation (CRP >5 mg/L and AGP ≤ 1 g/L)	57 (12)
Early convalescence (CRP > 5 mg/L and AGP > 1 g/L)	41 (8)
Late convalescence (CRP ≤ 5 mg/L and AGP > 1 g/L)	67 (14)
Inflammation-adjusted ferritin (µg/L) ^1^	25.9 (8.0–55.1)
Non-ID (ferritin ≥ 15 µg/L)	307 (63)
ID (ferritin < 15 µg/L)	185 (38)
Unadjusted ferritin (µg/L)	28.2 (9.3–62.9)
Non-ID (ferritin ≥ 30 µg/L)	238 (48)
ID (ferritin < 30 µg/L)	254 (52)
Non-ID (ferritin ≥ 15 µg/L)	314 (64)
ID (ferritin < 15 µg/L)	178 (36)
sTfR (mg/L)	7.5 (5.7–10.5)
Non-IDE (sTfR ≤ 8.3 mg/L)	288 (59)
IDE (sTfR > 8.3 mg/L)	204 (42)

AGP, α-1-acid glycoprotein; CRP, C-reactive protein; ID, iron deficiency; IDA, iron deficiency anaemia; sTfR, serum transferrin receptor; IDE, iron deficient erythropoiesis. ^1^ Ferritin values were adjusted for inflammation using the correction factors suggested by Thurnham et al. [30].

**Table 3 nutrients-12-00633-t003:** Diagnostic accuracy of the different Hb cut-off points to detect ID based on inflammation-adjusted and unadjusted ferritin, and iron and anaemia status in 18–25 year-old non-pregnant women of reproductive age residing in Soweto, South Africa (*n* = 492).

						Diagnostic Performance of Hb Cut-off Points to Detect ID		
	Total Anaemia (*n* [%])	IDA (n [%])	Anaemia without ID (*n* [%])	ID without Anaemia (*n* [%])	Non-ID & Non-Anaemic (*n* [%])	Sensitivity (%)	Specificity (%)	Youden Index
**Hb <12.00 g/dL** (As currently used in SA primary health care clinics to diagnose ID [inflammation-adjusted ferritin <15 µg/L])	91 (18.5)	62 (12.6)	29 (5.9)	123 (25.0)	278 (56.5)	35.1	88.6	0.24
**Hb <12.35 g/dL** (ROC-curve-determined to diagnose ID [inflammation-adjusted ferritin <15 µg/L])	183 (37.2)	103 (20.9)	80 (16.3)	82 (16.7)	227 (46.1)	55.7	73.9	0.30
**Hb <12.45 g/dL** (ROC-curve-determined to diagnose ID [unadjusted ferritin <30 µg/L])	192 (39.0)	131 (26.6)	61 (12.4)	132 (25.0)	177 (36.0)	51.6	74.4	0.26
**Hb <12.45 g/dL** (ROC-curve-determined to diagnose ID [unadjusted ferritin <15 µg/L])	192 (39.0)	100 (20.3)	92 (18.7)	72 (14.6)	228 (46.3)	56.2	70.7	0.27
**Hb <12.50 g/dL** (Altitude-adjusted based on WHO recommendations to diagnose ID [inflammation-adjusted ferritin <15 µg/L])	192 (39.0)	105 (21.3)	87 (17.7)	80 (16.3)	220 (44.7)	56.8	70.8	0.27

Hb, haemoglobin; ID, iron deficiency; ROC, receiver operation characteristics; SA, South Africa; WHO, World Health Organization.

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
