# Peer review of "Adjusting Haemoglobin Values for Altitude Maximizes Combined Sensitivity and Specificity to Detect Iron Deficiency among Women of Reproductive Age in Johannesburg, South Africa"

_nutrients, 2020, doi:10.3390/nu12030633_

Round 1
Reviewer 1 Report
The study by Silubonde et al describes adjustment of hemoglobin values to detect iron deficiency in a more sensitive and specific way in a cohort of women of reproductive age in Johannesburg, South Africa. The study has important public health aspects since it proposes a modification in the current method that is used to detect iron deficiency in South Africa. The cohort size is good and the recruitment criteria have been well described. My comments can be found below:
The authors state that majority of the population is of black African descent. Is there data already available that shows the variation in baseline hemoglobin values according to ethnicity? If yes (there is a high probability that it is the case), this should be included since this is another important factor that should be taken into consideration when thinking from a public health perspective. While the method suggested by the authors improve sensitivity, it does hamper the specificity as compared to the current healthcare practice (Table 3). The authors should comment on this.
Reviewer 2 Report
The paper addresses an important issue in the public health sector and its current policies on how to prevent/treat women with iron deficiency and highlights the diagnostic challenges. The paper demonstrates how important Hb adjustment for altitude is, in order not to miss women if simple Hb diagnostic is used. This is relevant for many other countries with extended regions at higher altitude and many primary health facilities with only limited diagnostic tools and therefore is relevant for the public health sector. The paper is very well and clearly written and structured. The results are well presented. A few additional points (as highlighted in the discussion part below) could be added to the discussion.
Abstract:
Line 18-19: The second sentence doesn’t make sense to me as I think once you detect low Hb this is the stage when you detect IDA, but it is difficult to define late stage IDA. I think the sentence should rather be late stage ID. And kind of indicating but at this stage it really needs to be detected if Hb is the only indicator. Line 26:’ The altitude adjusted threshold of 12.5..’, I think it would be good to add WHO here, so it is clear that this is the recommended method. I somehow miss a statement that your analysis support adjustment at the level as recommended by WHO. I think that would be important to mention in the abstract.
Introduction:
Line 72: I think this should say’…underestimation of the anaemia and IDA prevalence…’ not ID prevalence. If Hb is below the cut-off it is always IDA. Unless we can define a higher Hb cut-off that defines ID prior to having IDA. Line 104: was one aim also to compare to the recommended WHO cut-off?
Methods:
Line 110: Is SAMRC that well known that the abbreviation is fine to be used? It is the first time used in the text, it is only used in the author’s affiliation section but there also only the abbreviation is used. Line 118: why ‘...at least 3 months.’? Is this thought to be enough to adapt for the altitude? Line 124: how were the 2 clusters chosen? Are there differences in the 30 clusters? How big were the clusters (no of households)? Was the aim to enroll the same numbers from both clusters? Line 156: ‘A venous blood sample was…’ not ’A venous blood samples were…’ Line 174: You are stating the cut-off of the method used by Erhardt et al (8.3 mg?/L). However, when reading the Quansys publication by Esmaeili in 2019, they use a cut-off of 5.33. Wouldn’t you need to correct the sTfR values if you use a cut-off of 8.3? 8.3 comes from the RAMCO assay which was used to compare Erhardt’s method, but for comparing the Quansys method the Roche sTfR assay was used and thus a lower cut-off. Line 185: is this the correct cut-off to be used?
Results:
Line 199: You are using IQR here, while you use 25th-75th percentile in line 183. Whle this is the same it would be good to be consistent. Table 1 uses 25th-75th percentile again. Line 216: Looking at the high IDE prevalence of 42%, with a cut-off of 42% I am wondering what this would be if the cut-off described in the Sysmex publication would be used. Or, are you using Ramco-corrected values here and thus using 8.3 is fine? Line 251: ‘…based on an unadjusted ferritin…’ not ‘…based on unadjusted a ferritin…’; same in line 253
Discussion:
Line 272: I miss this statement in the abstract Lines 292 to 294: please check sentence Lines 297-301: You are comparing your results with the U.S. NHANES results where they get an optimal Hb cut-off of <12.8. Is this also for a population living around the same altitude above sea level? This is not very clear. 12.8 seems rather high unless they are living at a certain elevation. HemoCue 201+ was found to rather overestimate anemia prevalence in comparison to lab-based autoanalyzers while HemoCue 301 rather underestimates anemia prevalence. Lines 321-323: it might be worth to state which method was used to assess Hb in the two surveys. Was it also HemoCue 201+? Line 329: Did you also run your analysis using the BRINDA correction? Would be interesting to see whether it is comparable? Line 343: Maybe including a statement on the comparison of HemoCue 201+ with autonalayzers and capillary vs. venous blood would be worth. Thinking of HemoCue 201+ rather measuring lower Hb concentrations, this would suggest that if measured with an autoanalyzer from venous blood, the cut off for Hb would be slightly higher, and therefore getting even closer to the WHO recommended one. Line 345: Maybe a statement why you did not consider smoking would be good. How did Hb concentration between smokers and non-smokers compare? Or, if only a few smokers, would excluding them still give the same results? 348: sTfR performed quite poorly in comparison to Roche when measured with Quansys. Maybe something needs to be added here. Could the 22% self-reported HIV infections have influenced your findings? Could the high prevalence of overweight/obesity have had an impact on your results if we think about chronic inflammation in such a population?
Conclusions:
This section is well written and to the point.
Tables and Figures:
Table 1:
How was the cut-off for MUAC chosen? Do you have any reference that could be stated in line 143?
Author Response
Please see attachment below

Round 2
Reviewer 1 Report
The major concerns raised have been addressed by the authors and reflected by the changes made in the latest manuscript version.